# Actinic Keratoses (AK): An Exploratory Questionnaire-Based Study of Patients' Illness Perceptions

Dimitrios Sgouros [1,*,†], Adamantia Milia-Argyti [2,†], Dimitrios K. Arvanitis [1], Eleni Polychronaki [2], Fiori Kousta [2], Antonios Panagiotopoulos [2], Sofia Theotokoglou [1], Anna Syrmali [1], Konstantinos Theodoropoulos [1], Alexander Stratigos [2], Dimitrios Rigopoulos [2] and Alexander Katoulis [1]

[1] 2nd Department of Dermatology-Venereology, "Attikon" General University Hospital, Medical School, National and Kapodistrian University of Athens, 12462 Athens, Greece; arvandi18@yahoo.com (D.K.A.); theotokoglousofia@gmail.com (S.T.); annasyrmali@gmail.com (A.S.); theod28@gmail.com (K.T.); akatoulis@med.uoa.gr (A.K.)

[2] 1st Department of Dermatology-Venereology, Andreas Sygros Hospital, Medical School, National and Kapodistrian University of Athens, 16121 Athens, Greece; adamantia.milia.argyti@gmail.com (A.M.-A.); epolychronaki@yahoo.gr (E.P.); fkousta@gmail.com (F.K.); apanagiotopoulos2@gmail.com (A.P.); alstrat@med.uoa.gr (A.S.); drigopoul@med.uoa.gr (D.R.)

\* Correspondence: disgo79@gmail.com or dsgouros@med.uoa.gr; Tel.: +30-69-74816025 or +30-21-0583-2396; Fax: +30-21-0583-2396

† These authors contributed equally to this work.

**Simple Summary:** We recorded 208 patients receiving treatment for AK and conducted a cross-sectional questionnaire-based study, which aimed to investigate patients' perceptions of their illness. Our main objective was the detection not only of the illness perception of AK patients, but also of its influence on their perception of treatment and the correlation with patients' demographic characteristics and history, as well as the readiness to use sunscreen. The rising incidence of AK and its socioeconomic burden place the illness perception of AK patients among the most important barriers to overcome for the effective management of the disease. To the best of our knowledge, this is one of the first studies to attempt to unveil the illness perceptions of AK patients and their correlation with patients' demographics and sunscreen use and the influence on AK treatment. We strongly support reinforcing the awareness of AK and the role of dermatologists is crucial for this direction.

**Abstract:** Background: Decreased illness perception among actinic keratoses (AK) patients is a major barrier to the effective management of AK. Objective: We aimed to investigate patients' illness and treatment perceptions, their correlation to demographics and AK/skin cancer history, and secondarily the influence of these perspectives on treatment and sunscreen use. Materials and Methods: Participants completed questionnaires based on the Brief Illness Perception Questionnaire and statistical analysis was performed. Results: In total, 208 AK patients were enrolled. A large proportion were poorly aware of the disease (41.4%), with less than half (43%) being familiar with AK. Patients were aware of the chronic nature of the disease and its correlation to sunlight regardless of demographic characteristics. The level of education played a role in disease awareness ($p = 0.006$), and treatment plan perception ($p = 0.002$). The increase in sunscreen protection after AK diagnosis was higher in women ($p = 0.009$) and younger patients ($p = 0.044$). Patients' concerns regarding treatment were mainly related to the duration (30%) and effectivity (25%). Dermatologists' statements highlighting that AK are precancerous lesions (86.2%) influenced patients' willingness for treatment. Conclusion: Improved awareness of AK is necessary to increase treatment seeking and compliance, regarding both treatment and sunscreen use. Dermatologists' statements may have critical influence on patients' decisions to receive treatment for AK.

**Keywords:** actinic keratosis; non-melanoma skin cancer; illness perception; awareness; prevention





## 1. Introduction

Actinic keratoses (AK) are a very common skin disease that follows the chronic cumulative action of ultraviolet radiation. They are intraepidermal skin neoplasms, which correspond to the focal areas of irregular proliferation and differentiation of keratinocytes, with a low potential of malignant transformation to invasive squamous cell carcinoma (iSCC) [1,2]. They number among the commonest pre-neoplastic lesions and are considered to be a part of the evolutionary spectrum of SCC, constituting the most common neoplasm within the continuum of keratinocyte skin cancer, with a considerable impact on patients' quality of life [3]. Their prevalence, etiopathogenesis, risk factors, clinical presentation, dermatoscopic and histopathological features, and available treatments are well described in the current literature [4,5]. Their incidence, although poorly documented, is higher in the age range of 70 years and is expected to gradually increase, along with the ageing of the population [6,7]. Various treatments have been traditionally proposed to manage this condition, such as topical imiquimod, sodium-diclofenac, piroxicam, 5-fluorouracil, cryotherapy, photodynamic therapy, and surgery. Topical colchicine has been proposed to manage such conditions since the late 1960s [8].

Despite their high frequency, growing incidence and premalignant potential, AK are often underdiagnosed and therefore undertreated, which increases the burden of the disease and consequently that of non-melanoma skin cancers (NMSC) to patients and health care systems. The special chronic nature of this disease, with its indolent clinical behavior, the not always clear-cut perceived need for treatment and the underestimated role of sun protection, and the treatment barriers (such as cosmetic concerns, adverse events, compliance and treatment cost) are some of the main problems encountered in clinical practice and directly bear upon the poor awareness of the disease and the decreased perception of the potentially important consequences between patients if left untreated. The treatment of AK lesions relies on patients' perspectives regarding available treatment modalities and their investigations can correspondingly optimize adherence and clinical outcomes. Personal reported outcomes (PROs), directly from the patients, help in the investigation of AK treatment perceptions, especially because of the variety of the existing treatment options; instruments for their measurement should be incorporated in studies, as they are helpful for defining patients' preferences and beliefs [9].

Illness perception and its main domains are confirmed to determine patients' beliefs and therefore behavior, mainly regarding seeking medical care, and are related to the improvement of patients' outcomes [10–12]. Reliable tools for their investigation are the different versions of the Illness Perception Questionnaire (IPQ), which assesses patients' cognitive representations and emotional responses, revealing their understanding of the condition, physical or psychological impairment and its controllability [13].

Regarding AK, patients' willingness to seek and receive treatment depends on their perception of the disease, as a potentially malignant condition [14]. Previous studies have shown that information framing and the format of presentation seriously affects patients' intention to be treated and delineate compliance [14]. The successful treatment of AK should use a patient-centered plan which has a high efficacy with minimum side effects, also taking into consideration each patient's profile, including their perception of the disease, to improve adherence [15,16].

Our study's primary objectives were to investigate the perceptions of AK patients with regard to their illnesses (risk factors, chronicity, symptoms, and knowledge of the disease) and treatment (willingness to start treatment, understanding of treatment plan, and concerns) and the correlation to patients' demographics and related AK or skin cancer history. As a secondary goal, we detected the influence of these perceptions on treatment compliance and the readiness to use sunscreen.

## 2. Materials and Methods

This was an exploratory, cross-sectional, questionnaire-based study using self-administered paper-form structured questionnaires, conducted by the 1st and 2nd Dermatology De-

partments "A. Syggros" and "Attikon" University Hospitals (waiver decision by Ethics Committee 3555/4-3-2021). The only inclusion criterion was adult patients with AK, older than 18 years old, who were able to read independently of medical history, comorbidities, or disease severity.

The questions were based on previous research on AK and on the Brief IPQ model, due to its advantages of brevity and the speed of completion [17]. Patients were asked to complete a total of twenty, mostly close-ended, questions. The part of the questionnaire regarding demographics covered close-ended, one-response questions about gender, age (open-ended question), education level and history of UVR exposure and skin cancer. Questions about illness perception included both close-ended, multiple-choice questions of one or multiple responses and numerical ones on a 10-point rating scale.

Questions of the first part were used to evaluate illness perception, coherence and knowledge about the identity of the disease, source of information about AK, timeline and personal control, including emotional representation and concern. The causal representation of the disease was evaluated by a multiple-choice question, a variant of a question adopted from the IPQ-R version [18], followed by the prioritization of the three major etiological attributions, and sun exposure was evaluated as a causal factor with a 10-point scale rating scale response. Additional questions, beyond Brief IPQ, focused on sunscreen use before and after AK diagnosis, along with the readiness to increase sunscreen use.

The second part of the questionnaire concerned illness perception, coherence, controllability by treatment (control, consequences, and concern) and compliance. Regarding compliance, an extra question with two different scenarios was formulated, evaluating the impact of information framing by dermatologists on patients' intention to receive treatment for AK or not (S1. Questionnaire).

Patients' enrollment was voluntary. AK patients who attended the special Outpatient Dermato-Oncology Departments of the aforementioned hospitals between March and September 2021 were asked to complete the distributed questionnaires in written form, during their waiting time. All patients were first informed about the aim and method of the study, as well as the results being used exclusively for scientific reasons.

After data collection, a statistical analysis of the demographic factors was performed along with the questions of illness perception and perception of treatment.

*Statistical Analysis Method*

For continuous variables the mean, standard deviation and range or median, 25th and 75th percentiles and range were calculated after testing for normal distribution. Shapiro–Wilk and Shapiro–Francia tests were applied for normality testing. For categorical variables, the frequencies and percentages were calculated. Chi-squared and Fischer's exact tests were used for the comparison of categorical variables while the non-parametric Kruskal–Wallis and Mann–Whitney U tests were also applied. Spearman's Rho correlation coefficient was used to assess the strength of the relationship.

A $p$ value $< 0.05$ was considered statistically significant. All statistical analyses were performed using Stata/IC version 15.1.

## 3. Results

### 3.1. Patients' Demographics

A total of 230 self-report questionnaires were distributed to patients with AK. The response rate was 94.7%, with 218 questionnaires that were returned. To avoid non-response errors, 10 non-completely filled out questionnaires were excluded and our final sample comprises 208 patients.

Males prevailed in the group of patients with 168/208 (80.8%) members and the median age of the total of patients was 70 years. However, 78% of the patients were older than 65 years old, both male (77.78%) and female (78.95%) patients. Most patients had secondary education (40.9%, 85/208), while only one to three patients had completed higher education. Taking part in occupational sun exposure or outdoor activities was reported by

46.9% of the patients (97/208). A personal history of any type of skin cancer and solarium use, as independent risk factors, were not prevalent in the group of investigated patients. In total, 163/208 patients (79.1%) reported no history of any type of skin cancer and 194/208 (93.7%) reported no use of solarium ever in the past or at the time of investigation. All the results can be seen in Table 1.

**Table 1.** Patients' demographics and characteristics.

| Gender | *n* (%) |
|---|---|
| Males | 168 (80.77) |
| Females | 40 (19.23) |
| **Work outdoors/Outdoor activities** | |
| Yes | 97 (46.86) |
| No | 110 (53.14) |
| **Solarium** | |
| Yes | 13 (6.28) |
| No | 194 (93.72) |
| **Skin cancer history** | |
| Yes | 43 (20.87) |
| No | 163 (79.13) |
| **Level of education** | *n* (%) |
| Primary school | 34 (16.35) |
| High school | 85 (40.87) |
| University/technical studies | 66 (31.73) |
| MSc/PhD title | 23 (11.06) |

| Age | Median (25th–75th percentiles) | Range |
|---|---|---|
| All (*n* = 208) | 70 (65–76) | 34–88 |
| Males (*n* = 162) | 71 (65–76) | 40–88 |
| Females (*n* = 38) | 69 (65–75) | 34–87 |

*3.2. Illness Perception of AK*

Regarding the IPQ-R Illness perception dimensions, 119/208 patients were aware of the identity (title) of the disease (48.6%), compared to 84/208 patients (41.4%), who did not know the term AK. Most patients (63%) were informed only by their dermatologist about their disease, while clinicians of other specialties and the media were reported as a source of information at lower percentages (14.4% and 12%, respectively). A large proportion of investigated patients (45.9%, *N* = 94) knew that they had more than four lesions to be treated, but one in every three patients was not aware of the number of their AK lesions. Of note, 46.4% (*N* = 96) had already received treatment for AK more than four times, while 42.5% (*N* = 88) had done so at least once. Despite having multiple treatments, only 42% (*N* = 87) were familiar with the term AK; one in every three patients reported no familiarity with none of the questioned diseases (MM, BCC, SCC, AK) and 25% (*N* = 53) were more familiar with MM. At this point, it was deemed necessary to evaluate the percentage of patients with a history of skin cancer that was familiar with MM: beyond AK, only 24.5% (*N* = 13) were aware of MM. Regarding the duration of AK disease, patients had a median score of 3 years after the initial AK diagnosis (25th–75th percentiles 1–3; range: 0–26). All the aforementioned results can be seen in Table 2.

Among the illness perception dimensions, high median scores were gathered for timeline using the 10-point rating scale: chronicity was evaluated with a median score of 6 (25th–75th perc. 2–10; range: 1–10). Patients did not report intensive symptoms from AK (median score: 1; 25th–75th perc. 1–3; range: 1–10) and the feeling of coherence was rated as very good (median score: 8; 25th–75th perc. 3–10; range: 1–10). Emotional representation, investigated as concern, distress or anxiety provoked by AK, was evaluated as less than average (median score: 3; 25th–75th perc. 1–7; range: 1–10) and economic burden as minimal (median score: 1; 25th–75th perc. 1–3; range: 1–10).

**Table 2.** Illness perception questions, Questionnaire Part A1.

| Knowledge of the Term AK | *n* (%) |
|---|---|
| Yes | 119 (58.62) |
| No | 84 (41.38) |
| Information about AK except from dermatologist | *n* (%) * |
| Doctor of another specialty | 30 (14.42) |
| Internet/television | 25 (12.02) |
| Pharmacy | 7 (3.37) |
| Friends/other contacts | 9 (4.33) |
| Family | 14 (6.73) |
| None of the above | 131 (62.98) |
| Patients' familiarity with illnesses | *n* (%) * |
| MM | 53 (25.48) |
| BCC | 8 (3.85) |
| SCC | 8 (3.85) |
| AK | 87 (41.83) |
| None of the above | 62 (29.81) |
| Familiarity with MM Skin Cancer history | *n* (%) |
| Yes | 13 (24.53) |
| No | 40 (75.47) |
| How Many AK do You Know That You Have? | *n* (%) |
| 2–3 | 50 (24.39) |
| 4+ | 94 (45.85) |
| I do not know | 61 (29.76) |
| How many times did you have already treatment for AK? | *n* (%) |
| 0 | 23 (11.11) |
| 1–3 | 88 (42.51) |
| 4+ | 96 (46.38) |

* They do not add up to 208 patients &100%, because some patients gave more than one choice.

Sunlight was highly assessed as a causal factor and most patients blamed it absolutely for their disease (median score: 10; 25th–75th perc. 8–10; range: 1–10). Before AK diagnosis, sunscreen use was poorly reported (median score: 1; 25th–75th perc. 1–5; range: 1–10), while after AK diagnosis the reported use or willingness to increase sunscreen use was very high (median score: 10; 25th–75th perc. 7–10; range: 1–10). These results can be seen in Tables 3 and S1.

**Table 3.** Illness perception questions based on Brief IPQ model, Questionnaire Part A2.

| | Median (25th–75th Percentiles) | Range |
|---|---|---|
| How long do you think your illness will last? | 6 (2–10) | 1–10 |
| How much do you experience symptoms from your illness (e.g., pain, itch)? | 1 (1–3) | 1–10 |
| How well do you feel you understand your illness? | 8 (3–10) | 1–10 |
| How much does your illness affect you (e.g., does it make you angry, scared, upset, depressed)? | 3 (1–7) | 1–10 |
| How much do you think solar radiation is responsible for your illness? | 10 (8–10) | 1–10 |
| How often did you use sunscreen BEFORE your AK diagnosis? | 1 (1–5) | 1–10 |
| How often do you use / how willing are you to use sunscreen AFTER your AK diagnosis? | 10 (7–0) | 1–10 |
| How much do you feel your illness affects (burdens) you financially? | 1 (1–3) | 1–10 |

Among causal attribution dimensions, after the prioritization of the three main factors blamed to be responsible for AK, solar radiation, at 79.9% (*n* = 163), was prominently displayed. In total, 40% (*n* = 64) considered ageing as the second most important factor, followed by stress (21.3%, *n* = 34) and pollution (19.6%, *n* = 26), while the third most important cause was also ageing (24.1%, *n* = 32), followed by pollution (19.6%, *n* = 26), stress and accident, each at 12% (*n* = 16).

A possible relation of the causal representation of AK with gender, education level and history of skin cancer was also investigated. Male patients blamed serially solar radiation, ageing and pollution with 81.1% (*n* = 133), 44.2% (*n* = 57) and 20.4% (*n* = 21), respectively. Most female patients (75%, *n* = 30) also blamed solar radiation, and rated stress as the second (32.3%, *n* = 10) and ageing as the third (43.3%, *n* = 13) most important causal factors and did not considerably prioritize pollution or accident (Table S3).

No differentiation related to education level was noticed, with patients of all education levels indicating that sunlight and ageing were the most important factors (Table S4). Patients with no skin cancer history prioritized the same main causal factors, likewise with patients with a history of skin cancer as well, with the latter blaming solar radiation more (91% with history; 77% with no history) (Table S5).

### 3.3. Information Framing—AK Treatment

Most patients (86.2%, *n* = 175) were likely to want treatment after hearing their dermatologists' statements that AK are precancers. Equally, 72.7% (*n* = 133) would decide to receive treatment for AK upon hearing the statement that about 0.5% of AK will turn into skin cancer. Regarding concerns related to treatment, 35.2% (*n* = 72) had no concerns, followed by 30% (*n* = 62) which worried mostly about treatment duration. Treatment efficacy was the main worry for 24.6% (*n* = 51) of patients, while cost was not a preoccupation for the patients of our study (Table 4). Possible differentiations based on gender were also investigated. A total of 37% of males had no concern with regard to treatment (*n* = 62), treatment duration and effectiveness; 32% (*n* = 53) and 22% (*n* = 37), 9% (*n* = 15) were preoccupied with safety, while cosmetic outcomes and pain worried 5.4% (*n* = 9) and 4.8% (*n* = 8) of male patients, respectively. Females, in comparison, were more concerned about the effectiveness of AK treatment (36%, *n* = 14), cosmetic outcome and pain (20.5%, *n* = 8, 15.4%, *n* = 6). The percentage of female patients with no concerns about treatment was similar to the male ones though (28%, *n* = 11) (Table S6).

**Table 4.** Treatment perception questions, Questionnaire Part B.

| AK Are Precancers | *n* (%) |
| --- | --- |
| Yes | 175 (86.21) |
| No | 28 (13.79) |
| About 0.5% of AK will turn into skin cancer | |
| Yes | 133 (72.68) |
| No | 50 (27.32) |
| Concerns about treatment | *n* (%) |
| Duration | 62 (29.95) |
| Safety | 19 (9.18) |
| Efficacy | 51 (24.64) |
| Pain/skin reaction | 14 (6.76) |
| Cosmetic outcome | 17 (8.21) |
| Cost | 1 (0.48) |
| None of the above | 73 (35.27) |

They do not add up to 208 patients or 100% because some patients gave more than one answer.

For the rest questions regarding AK treatment perceptions, understanding of the treatment plan was highly rated (median score: 10; 25th–75th perc. 8–10; range: 1–10), was reported compliance to the proposed treatment (median score: 10; 25th–75th perc. 8–10;

range: 1–10). Illness control by the followed treatment was significantly highly rated too (median score: 10; 25th–75th perc. 9–10; range: 1–10) (Tables 4 and S7).

### 3.4. Correlations

Statistically higher was the relation of gender with the knowledge of the number of AK lesions ($p = 0.002$), with female patients not knowing the number of their lesions being 15.4% against 33.1% of males, even though half of the male patients had already had at least four treatments, as compared with 33% of females ($p = 0.03$); emotional representation was higher in women ($p = 0.048$), as was sunscreen use before AK diagnosis ($p < 0.001$). Willingness to increase sunscreen use after AK diagnosis was determined to be equally high for both genders; the 25th–75th percentiles were wider in male patients though ($p = 0.009$). Financial burden related to treatment was reported only by females ($p = 0.03$, $n = 1$) (Tables 5 and S8a).

**Table 5.** Correlations of illness and treatment perception questions with patients' demographics and characteristics.

| | Gender (*p*-Value) | Outdoor Work/Activities (*p*-Value) | Solarium (*p*-Value) | Skin Cancer History (*p*-Value) | Level of Education (*p*-Value) | Age (*p*-Value) | Age $\geq$ 65 (*p*-Value) |
|---|---|---|---|---|---|---|---|
| Do you know the term AK? | 0.843 | 0.145 | 0.246 | **0.037** | **0.019** | **0.023** | 0.657 |
| Years from AK diagnosis | 0.908 | 0.478 | 0.709 | **<0.001** | 0.55 | 0.143 | 0.132 |
| Number of AK lesions | **0.002** | 0.334 | 0.467 | 0.249 | 0.94 | **0.025** | 0.159 |
| Number of AK treatments | **0.03** | *0.052* | 0.776 | 0.223 | 0.202 | 0.236 | *0.086* |
| How long do you think your illness will last? | 0.607 | 0.104 | 0.462 | **0.002** | 0.14 | 0.788 | 0.562 |
| How much do you experience symptoms from your illness? | 0.602 | 0.393 | 0.764 | 0.34 | 0.34 | 0.586 | 0.446 |
| How well do you feel you understand your disease? | 0.421 | 0.21 | 0.186 | *0.054* | **0.006** | *0.081* | 0.693 |
| How much does your illness affect you? | **0.048** | 0.383 | 0.988 | 0.857 | 0.431 | 0.257 | 0.558 |
| How much do you think solar radiation is responsible for your illness? | 0.123 | 0.193 | 0.616 | *0.088* | 0.688 | 0.472 | 0.715 |
| Sunscreen use *before* AK diagnosis | **<0.001** | *0.083* | *0.09* | 0.236 | **<0.001** | **0.017** | *0.058* |
| Sunscreen use *after* AK diagnosis | **0.009** | 0.912 | 0.574 | 0.747 | 0.288 | **0.044** | 0.553 |
| Financial burden | **0.03** | 0.222 | *0.071* | 0.334 | **0.033** | *0.072* | 0.165 |
| Treatment if AK precancerous | 0.305 | **0.014** | 0.697 | 0.457 | 0.148 | 0.186 | 0.333 |
| Treatment, if about 0.5% will turn into skin cancer | 0.418 | 0.217 | 0.754 | **0.047** | 0.198 | 0.424 | 0.137 |
| Perception of treatment plan | 0.455 | 0.896 | 0.139 | 0.649 | **0.002** | 0.617 | 0.682 |
| Ease to comply with suggested treatment | 0.477 | 0.112 | *0.061* | 0.402 | **0.039** | 0.899 | 0.679 |
| Feeling of control of the disease by treatment | 0.857 | 0.704 | *0.054* | 0.239 | 0.149 | 0.393 | 0.792 |

Bold-faced *p*-values indicate statistical significance ($p < 0.05$); italics-faced ones indicate the tendency to display statistical significance.

Outdoor activities and solarium use had no statistical significance with the quantitative variables. Patients with a history of occupational UV exposure reported multiple AK treatments ($p = 0.052$) and were more likely to receive treatment when AK were described as precancerous lesions ($p = 0.014$), while patients with no such history reported higher sunscreen use before AK diagnosis ($p = 0.083$) (Tables 5 and S8b,c).

Skin cancer history was found to be statistically significantly with regard to the knowledge of the term AK ($p = 0.037$), the mean years of AK diagnosis ($p < 0.001$) and disease duration ($p = 0.002$). Patients with skin cancer history were also more likely to receive treatment when informed that 0.5% of AK will turn into skin cancer ($p = 0.047$) and tend to have better coherence, without statistical significance though ($p = 0.054$) (Tables 5 and S8d).

The parameter of knowledge of disease identity ($p = 0.019$) and coherence ($p = 0.006$) was also statistically high in patients with higher education levels. Similarly, patients with higher education were determined to have a better perception of treatment plan ($p = 0.002$) and better compliance ($p = 0.039$) and reported higher sunscreen use before AK diagnosis ($p < 0.001$). Contrarily, no statistical significance was ascertained in relation to sunscreen use

or willingness to increase it after AK diagnosis ($p = 0.288$) in patients with higher education levels (Tables 5 and S8e).

From the correlation of age with illness and treatment perception questions, statistically significant was the relation of knowledge of disease identity in younger patients ($p = 0.039$). Older patients were less aware of their number of AK lesions ($p = 0.025$), reported lower sunscreen use before AK diagnosis ($p = 0.017$) and indicated a significantly lower readiness to increase sunscreen use after their diagnosis ($p = 0.044$). In older patients, a tendency toward reduced coherence and increased financial burden was found, although this was not statistically significant ($p = 0.081$, $p = 0.072$, respectively). For patients older than 65 years, who were the majority of our sample, no statistically significant correlations were demonstrated, but 50% reported multiple treatments, in comparison to 32% of patients who were younger than 65 years old, and there was a lower median score for sunscreen use before AK diagnosis too. All the results are shown in Tables 5 and S8.

Those patients that were aware of the title their disease also had a better perception of illness duration ($p = 0.004$), coherence ($p < 0.001$) and the suggested treatment plan ($p = 0.044$) (Table S9). Among correlations of the questions regarding information framing, the patients that were likely to receive treatment in one of or both of the different scenarios were found to be more concerned, stressed or depressed about their disease ($p = 0.042$), and had a better perception of the treatment plan ($p = 0.040$). Furthermore, a tendency to feeling better towards disease control by treatment was demonstrated in this group ($p = 0.087$); however, no better reported compliance was determined ($p = 0.123$) (Tables S9 and S10).

## 4. Discussion

### 4.1. Illness Perception of AK Patients

There are very few and heterogenous reports investigating illness perceptions of AK patients. Available literature data include only one study of 200 Turkish patients [19] and one of 2400 patients from western Saudi Arabia [20], and the present one is the first attempt, to the best of our knowledge, to investigate the perceptions of AK patients in the European population.

Considering that most investigated patients (almost 90%) had received treatment for AK at least once and that the median rate was 3 years after their initial diagnosis, the results show that patients do not have good knowledge about the identity of their disease: 58.6% knew the title of the disease and one in every three patients did not know their number of AK lesions, indicating low coherence, despite the chronicity of their disease. All in all, this could be related to the poor symptomatology of AK [21,22], which also aligns with the reported rating, and could potentially be an additional barrier, leading to the underestimation of AK, especially regarding consequences, concerns and emotional representation. The psychological impairment of AK patients is reported in the literature, regarding the impacts on quality of life, self-confidence and well-being in general [3,23]. These are related not only to concern about disease evolution, but also to behavioral recommendations, such as direct sun exposure [24].

Even though AK are one of the commonest skin diseases globally, the nature of the disease is widely unknown: awareness of AK among the general population of some European countries, Australia and the USA reaches 6–7% on average [23,25,26]. According to Halpern et al. (2005), there is more awareness and knowledge of a disease in areas of higher incidence. Indeed, in Australia, where NMSC incidence reaches up to 25%, and in the USA (10%), in comparison to <3% of Europe, patients reported better familiarity with BCC than with AK (30% vs. 7%, respectively) [25]. Contrarily, 10% of patients in Saudi Arabia knew BCC; only 1% knew of AK and 8.6% both [20]. In our study, with 80% of the patients not having skin cancer history, one out of three patients reported no familiarity with AK, BCC, SCC, and MM, and only one out of four knew MM, even with skin cancer history. Despite multiple reported treatments, less than half of the patients were familiar with the term AK, further supporting the low overall knowledge of disease identity. The investigation of the familiarity with MM of 25.5% of our patients would be of interest, to

define a possible relation with the known higher risk of MM or better information at the state or physician side. [26]

In the aforementioned study of Halpern et al., most AK patients (66%) listed media as their main source of information about the disease, as compared with 12.4% in our study. In the latter, 65% of patients were informed only by attending dermatologists, like in the study of Basyouni et al., while physicians of other specialties were chosen by 15%. In countries with better knowledge and awareness of skin cancer, such as the USA or Australia, the media plays a significant role in public awareness [20], which could be inferentially reinforced. Further, dermatologists should raise awareness not only of treated patients, but also of other physicians to avoid underdiagnosis and increase appropriate medical advice seeking. Clear communication of information about AK, even in written form, could increase the understanding of this disease. Even in the skin cancer-related apps era, clinical consultation outclasses the media, i.e., printed or television [27]. Dermatologists are mainly responsible for the increase in disease awareness, in the direction of knowledge of skin health, increase in sun protection behaviors and decrease in skin cancer incidence [28].

In our study, dimensions of coherence and disease duration were highly rated, despite the poor knowledge of disease identity. Patients aware of the disease title had better perception of the chronicity ($p = 0.004$), coherence ($p < 0.001$) and treatment plan ($p = 0.044$), while skin cancer history was significantly related only to increased perception of disease duration ($p = 0.002$). In a like manner, in Akarsu et al.'s study, patients with skin cancer history had a better perception of the disease's chronicity, including men [19]. Skin cancer history was also found to be related to an increased negative perception of consequences and controllability of AK by treatment, which was not confirmed in our study. In the latter, gender was also found to play no role in the timeline perception of the disease.

On the other hand, the level of education was shown to be an important factor for increased coherence and sunscreen use. Patients with higher education surpass primary school graduates with regard to their understanding of the disease ($p = 0.006$). After AK diagnosis, sunscreen use or readiness for increased use was equally high among patients, in contrast to previous studies [19,29], where it was higher in higher education levels. This could be explained by the fact that patients of lower education levels tend to attribute their disease to psychological or accidental factors, something that was not proven in our study. Despite the similar willingness of patients with lower education levels to increase sunscreen use, treatment plan perception was higher among patients with higher education levels ($p = 0.002$), who also had better outcomes regarding compliance to the suggested treatment ($p = 0.039$).

Based on previous reports, the readiness to increase sun protection behavior is associated with gender and age [19,25,26,29–31]. Women tend to worry more about changes in their appearance and are more affected by AK diagnosis; they show, therefore, better adherence to treatment recommendations [19,25,26,32]. Emotional impairment also seems to be higher in women than in men, according to other studies of other skin diseases too [33–35]. Changes in appearance are of great importance to AK patients, and especially when located on the face, they lead to a psychological burden, which should not be underestimated [36]. Outcomes regarding emotional representation in our study, did not confirm the above-mentioned observations. Concern was overall rated lowly; it was positively associated with gender and knowledge of the disease identity. Female patients were found to have greater psychological ($p = 0.048$) and financial ($p = 0.030$) burdens after AK diagnosis in comparison to male ones, like in Akarsu et al.'s study, but contrarily were shown to use more sunscreen than men before ($p < 0.001$) and to be more willing to increase sunscreen use after diagnosis ($p = 0.009$), not equally with males [19].

Sunscreen use was nevertheless negatively associated with a higher age: the older the patients, the less sunscreen use reported before ($p = 0.017$) and after ($p = 0.044$) AK diagnosis, even though average overall readiness was high, independently of age and other demographics. Before AK diagnosis, sunscreen use was rated as rare, something that has also been observed in countries with increased public awareness towards NMSC [20,26].

Skin cancer history was not found to differentiate the perception of etiological attributions though, as found in previous studies.

Exposure to solar radiation was blamed clearly as the first cause of AK by 80% of patients, who have a good perception of the causative factors overall. Stress was prioritized as the second factor by patients of both genders, 21%. Gender and a lower level of education seemed to play a role in the attribution of the disease to psychological causes in the available literature data [19]. In our study, besides female gender, stress was prioritized more highly, which was not statistically significant, by patients with higher education levels.

### 4.2. Information Framing—Treatment Perceptions of AK Patients

Previous studies have already demonstrated the impact of information framing about a disease on a patient's intention to be treated [37–40]. Regarding AK, patients are more likely to choose treatment when the disease is presented as precancerous, a common expression used by the attending clinicians to describe AK. When the chances of a positive outcome are underlined, a lower proportion of patients are willing to receive treatment [39]. Changes in patients' responses about treatment according to the presentation of the information about AK emphasizes the importance of clinicians' statements. The framing of information about the disease has been shown to be more important than age, gender or skin cancer history, but most patients would choose to receive treatment anyway [39].

Likewise, in our study, 9 out of 10 patients were willing to receive treatment if AK was presented as a precancerous condition. With a lower but comparable percentage (73%), patients would receive treatment if their dermatologist stated that the risk for AK turning into skin cancer is about 0.5%. Patients with positive responses in at least one scenario were found to have more concerns about AK ($p = 0.042$) in any relation, regardless of general demographics or education level. Those who wanted treatment, even if informed of the risk of transformation of an AK being lower than 1%, had better treatment plan perception ($p = 0.04$) and a higher feeling of disease control by treatment ($p = 0.087$). AK are not considered a life-threatening disease, but seeking medical treatment for them is crucial. Using the appropriate terminology, adapted to each patient's knowledge and educational background, as part of a patient-centered disease management is essential because it determines patients' decision making [40].

Emphasis should be placed not only on the need for treatment, but also on the therapeutic options available, with discussions of patients' expectations, regarding efficacy, adverse events, cost and cosmetic outcomes, improving compliance [19,22,41]. Compliance is directly connected with treatment and illness perceptions in general, but is also associated with treatment duration, convenience, associated concerns and cost. AK patients tend to poorly comply with suggested treatments, as described in an observational study of Shergill et al., with percentages of non-compliance reaching up to 88% [41]. The duration of treatment is the main decisive factor. Patient-applied, shorter regimens, requiring less frequent application, are related to better adherence rates [42]. A shorter duration and better tolerance influence patients' perceptions of treatment outcome and therefore satisfaction and adherence; even if a treatment is inferior in terms of effectiveness, it is preferred by patients when better tolerable. [11,43]

In our study, AK patients were shown to perceive treatment quite successfully and have no difficulty following the treatment. Of note, due to the use of the self-report method, there is possibly a non-factual, increased feeling of adherence and therefore disease control by treatment, which does not correspond to the real monitoring, as also described in other studies of AK [43,44].

Not to disregard this, the overriding majority of patients were treated with cryosurgery and only two patients had undergone PDT. This could have had an influence on the answers regarding compliance, as each treatment option was accompanied by a different duration, application and adverse events profile, determining factors of adherence and therapy persistence. Surprisingly, the median rates of disease control by treatment perceptions were very high, even if 46% of patients had received multiple, more than four, treatments already.

In total, 35% of patients had no concerns regarding treatment parameters; the duration and efficacy of treatment are a matter of concern for 30% and 25% of patients though. Pain, skin reaction and cosmetic outcome were rated as lower in importance (7% and 8%, respectively). Cryosurgery can often lead to hypopigmentation or scar formation, which justifies cosmetic outcome concerns [45]. PDT on the other hand, is more connected with pain and local skin reactions [46]. These adverse events are considerable treatment barriers and influence patients' daily activities or social lives, especially when AK are located on visible sites, such as the face or hands [22,31]. PDT is found to have equivalent or even better recovery times and cosmetic outcomes and is often overall preferred for AK patients [47]. In our study, these adverse events were not found to have a considerable impact; the dominance of male gender in our sample could be a reasonable explanation, as females are mostly concerned about them, according to the commonest AK patients' profiles. The classification of AK patients in different profiles allows for the better management of the disease; patients of our study fall rather into the category of unengaged (with low medical engagement) [48].

The cost of treatment is another important factor affecting treatment adherence, the cost of daily sunscreen use included [16]. Previous studies support the idea that the cost related to NMSC is significantly higher than AK treatment- and sunscreen-related costs [28,49] and is therefore important not only to treat AK, but also investigate the impact of cost-related perceptions on compliance. Only one patient in our study was concerned about cost, and financial burden was overall rated as minimal. This can be related to the fact that patients in public hospitals do not have to pay for their treatment and any medication is covered by public health insurance. Further, sunscreen use was limited and possibly not considered as part of treatment cost. In conclusion, educating patients on the meaning of treatment cost is essential and should include both the appropriate dosage of medication to ensure efficacy and daily sunscreen use, as the basis of each treatment approach. Cost should not be a barrier to patients' willingness to increase sunscreen use, which in our study was high.

*4.3. Limitations*

Our study is the first one trying to investigate the illness perceptions of AK patients and the connection between demographic characteristics, sunscreen use and treatment in a large sample of Greek patients. However, it has certain limitations: it is a cross-sectional study, rather than a longitudinal one, and so long-term conclusions cannot be drawn. Further, the participants were patients from special departments of two tertiary hospitals, who already had an AK diagnosis and were, therefore, not naïve with regard to the disease, and represent only a proportion of the Greek AK population. A study of a wider scale with a bigger sample of patients, eventually also over a longer period, would be of interest. Sampling was based on patients' availability and a big part of different AK patient profiles may be underrepresented. It would be, therefore, wise to also include patients that receive other treatment modalities or emphasize inclusion criteria more precisely in future studies. All collected data were based on the self-report method and, considering the median age of participants, recalled information may be subjective. Finally, external validity may be restricted and results cannot be generalized due to the risk of sampling bias. Studies using personal interviews are suggested to overcome this risk.

**5. Conclusions**

In conclusion, the illness perception of AK patients is low, even in patients who receive treatment regularly. Despite the long duration of AK diagnosis, patients do not perceive the disease's identity well. Familiarity with skin cancers and AK is overall low, even in patients with skin cancer history. Knowledge about AK comes mostly from seeing dermatologists, while doctors of other specialties and the media contribute less to patients' information. Patients feel they understand their condition well and level of education seems to play a role here. The chronicity and timeline of AK are well perceived. Despite decreased coherence, AK patients feel that they are helped by treatment and report no difficulty

in adherence, with low levels of reported concerns. Overall, they evaluated control by treatment as greater than the personal control of the disease. Concern about AK is below average, while concern about treatment, regardless of gender, pertains to duration and efficacy. The readiness to follow a suggested treatment and increase sunscreen use is at a high level. Although sunlight seen as clearly responsible for AK, younger patients are more likely to use sunscreen and 15% use no sunscreen regardless of age, education level and skin cancer history. Demographic characteristics and skin cancer history are associated with AK illness perceptions, but do not differentiate the understanding of causative factors. Dermatologists are proved to form patients' illness perceptions. Not only are they the main source of information about AK, but also their wording affects patients' decisions to receive treatment. Increased awareness of AK is needed for the successful management of this disease. To the best of our knowledge, this is one of the first attempts to investigate the illness perceptions of AK patients. We strongly support the need for further studies to unveil important factors which impact AK illness perceptions.

**Supplementary Materials:** The following supporting information can be downloaded at: https://www.mdpi.com/article/10.3390/curroncol29070408/s1: S1. Questionnaire. Table S1: Illness perception dimensions. Table S2: Treatment perception dimensions. Table S3: Treatment concerns according to gender. Table S4: Prioritization of causal attributions. Table S5: Causal attributions according to gender n (%). Table S6: Causal attributions according to educational level n (%). Table S7: Causal attributions according to skin cancer history n (%). Table S8: Statistically significant correlations. Table S9: Statistically significant correlations of knowledge of the term AK. Table S10: Statistically significant correlations of readiness for treatment, if positive in one of the two questions of information framing.

**Author Contributions:** Conceptualization, D.S., A.M.-A., D.R. and A.K.; formal analysis, D.S. and A.M.-A., Investigation, D.S., A.M.-A., E.P., F.K., A.P., S.T., A.S. (Anna Syrmali), K.T., A.S. (Alexander Stratigos), D.R. and A.K.; methodology, D.S., A.M.-A., D.K.A. and A.K.; project administration, D.S., A.M.-A., D.R. and A.K.; supervision; D.S., E.P., A.S. (Alexander Stratigos), D.R. and A.K.; writing—original draft; A.M.-A. and D.S.; writing—review and editing; D.S., A.M.-A. and A.K. All authors have read and agreed to the published version of the manuscript.

**Funding:** This research received no external funding.

**Institutional Review Board Statement:** This study was conducted according to the guidelines of the Declaration of Helsinki and approved by the Ethics Committee of "A.Syggros" and "ATTIKON" University Hospitals of Athens (3555/4-3-2021).

**Informed Consent Statement:** Informed consent was obtained from all subjects involved in the study.

**Data Availability Statement:** The data presented in this study are available on request from the corresponding author.

**Conflicts of Interest:** The authors declare no conflict of interest.

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
