# Peer review of "Actinic Keratoses (AK): An Exploratory Questionnaire-Based Study of Patients’ Illness Perceptions"

_curroncol, doi:10.3390/curroncol29070408_

Round 1
Reviewer 1 Report
I read with great interest the original article titled "Actinic keratoses (AK): An exploratory questionnaire-based 2 study of patients’ illness perceptions" by Sgouros.
The article is a valid example of proof of concept that detect the disease perception in patients with AK
I will add only a paragraph to address the main study limitations before conclusions and a mention toward the state of art of PROs in AK [10.1111/dth.14833]
Reviewer 2 Report
A questionnaire exploring actinic keratosis illness perception among patients, reporting results similar to other studies. the main limitation of this paper is surely the low number of participants for the questionnaire.
However, I found the questionnaire well structured, and different findings are statistically significant;
Do I think the paper may be eligible to be published after some adjustments:
line 53 you should add: "Various treatments have been traditionally proposed to manage this condition, such as topical imiquimod, sodium-diclofenac, piroxicam, 5-fluorouracil, cryotherapy, photodynamic therapy, and surgery. Topical colchicine was proposed to manage such conditions since the late 1960s" and cite: doi: 10.3390/pharmaceutics14020294.
line 54-62 needs some referrals, such as: DOI: 10.23736/S2784-8671.20.06600-6
Thank You
Reviewer 3 Report
Actinic keratoses (AK): An exploratory questionnaire-based study of patients’ illness perceptions
The findings are interesting, but the presentation of the data can be improved. Pls see my comment below:
Abstract
Objective: To investigate and correlate the AK patients’ perceptions and their influence on treatment and readiness to use sunscreen
Introduction: Our study had as primary objectives to investigate perception of AK patients of their 77 illness (risk factors, chronicity, symptoms, knowledge of the disease) and treatment (will-78 ingness to start treatment, understanding of treatment plan, concerns) and its correlation 79 to patients’ demographics and related AK or skin cancer history. As a secondary goal, we 80 detected the influence of these perceptions on treatment compliance and the readiness to 81 use sunscreen.
COMMENT: Pls align the indicated objectives
COMMENT:
The only inclusion criterion was patients with AK, independent of medical history, comorbidities, or disease severity.
Patients’ enrollment was voluntary. AK patients who attended the special Outpatient 109 Dermato-Oncology Departments of the aforementioned hospitals between March and 110 September 2021, were asked to complete the distributed questionnaires in written form, 111 during their waiting time. All patients were first informed about the aim and method of 112 the study, as well as the use of the results exclusively for scientific reasons
COMMENT: The inclusion criterio should include adult more than 18 years old, those able to read and write in the
Materials & Methods: In total, 208 AK patients were enrolled. Participants completed questionnaires based on Brief Illness Perception Questionnaire and statistical analysis was performed
3.1. Patients’ Demographics
A total of 230 self-report questionnaires were distributed to patients with AK. Re-128 sponse rate was 94,7%, with 218 questionnaires that were returned. To avoid non-response 129 errors, 10 non-completely filled out questionnaires were excluded and our final sample 130 comprises 208 patients
COMMENT: The enrolment of participants should be placed at the Results section, not the method section.
Table 1. Patients' Demographics & Characteristics
Table 2. Illness perception questions, Questionnaire’s Part A1.
Table 3. Illness perception questions based on Brief IPQ model, Questionnaire’s Part A2.
Table 4. Treatment perception questions, Questionnaire’s Part B.
COMMENT: Table 1-4 need to be restructured. The descriptors should be listed in the same column
Table 5. Correlations of illness and treatment perception questions with patients’ demographics and 253 characteristics
COMMENT: this table needs modification-the presentation does not align with the correct way of presenting correlation
3.3. Information framing – AK treatment
3.4. Correlations
3.3 and 3,4= need to improve the clarify
4. Discussion
4.1. Illness perception of AK patients
4.2. Information framing - Treatment perceptions of AK patients
COMMENT: pls relate with more recent study reporting this : for instance :
https://www.tandfonline.com/doi/abs/10.1080/14764170802056117
https://www.ncbi.nlm.nih.gov/pmc/articles/PMC5439540/
https://www.sciencedirect.com/science/article/pii/S019096221201064X
4.3. Limitations 426
Our study is the first one trying to investigate illness perceptions of AK patients and 427 the connection between demographic characteristics, sunscreen use and treatment in a 428 large sample of Greek patients. However, it has certain limitations: it is rather a cross-429 sectional study, than a longitudinal one and long-term conclusions cannot be drawn. Fur-430 ther, the participants were patients from special departments of two tertiary hospitals, 431 who already had an AK diagnosis and were, therefore, not naïve with the disease, and 432 represent only a proportion of Greek AK population. Sampling was based on patients’ 433 availability and a big part of different AK patient profiles may be underrepresented. All 434 collected data were based on self-report method and considering median age of partici-435 pants recalled information may be subjective. Finally, external validity may be restricted 436 and results cannot be generalized due to risk for sampling bias.
COMMENT:Pls include the how these limitations can be overcome.
5. Conclusions 438
In conclusion, illness perception of AK patients is low, even in patients who receive 439 treatment regularly. About 60% perceives disease’s identity, while 30% does not know the 440 number of lesions to be treated, despite AK diagnosis duration over than 3 years. Famili-441 arity with skin cancers and AK is overall low, although 20% of patients reported skin can-442 cer history. Knowledge about AK comes mostly (65%) from attending dermatologist, 443 while doctors of other specialties and the media contribute less to patients’ information. 444 Patients feel they understand well their condition and level of education seems to play a 445 role here. Chronicity and timeline of AK are well perceived. Despite decreased coherence, 446 AK patients feel that are helped by treatment and report no difficulty in adherence, while 447 30% has no concerns about it. They overall evaluate control by treatment as greater than 448 personal control of the disease. Concern about AK is below average, while concern about 449 treatment, regardless of gender, pertains to duration and efficacy. Cost does not worry the 450 patients of our study. The readiness to follow suggested treatment and increase sunscreen 451 use is at high level. Although sunlight is blamed clearly as responsible for AK, younger 452 patients are more likely to use sunscreen and 15% uses no sunscreen regardless of age, 453 education level and skin cancer history. Women tend to intent on complying better with 454 sunscreen use after diagnosis. Demographic characteristics and skin cancer history are 455 associated with AK illness perceptions, do not differentiate understanding of causative 456 factors, though. Dermatologists are proved to form patients’ illness perceptions. Not only 457 they are the main source of information about AK, but also their wording affects patients’ 458 decision to receive treatment. In specific, 86% would receive treatment if AK presented as 459 precancers and 73% if they knew the risk of malignant transformation is about 0.5%. These 460 patients tend to understand suggested treatment plan better and have increased feeling 461 of AK control by treatment, however, they do not have better compliance. Increased 462 awareness towards AK is needed to the direction of successful management of the disease. 463 To the best of our knowledge, this is one of the first attempts to investigate illness percep-464 tions of AK patients. We strongly support the need for further studies to unveil important 465 factors with impact on AK illness perceptions.
COMMENT: pls summarise better the main findings.
Round 2
Reviewer 3 Report
The authors have made some changes to the text, but the format of the table / how data were results (Table 1 and Table 4) are incorrect still. Pls refer to other published papers.
Author Response
Please see the attachment

This manuscript is a resubmission of an earlier submission. The following is a list of the peer review reports and author responses from that submission.